# Bistatic Radar Observations Correlation of LEO Satellites Considering $J_2$ Perturbation

**Zongbo Huyan, Yu Jiang \*** **, Hengnian Li, Pengbin Ma and Dapeng Zhang**

State Key Laboratory of Astronautic Dynamics, Xianning Road No. 462, Xi'an 710043, China;
huyanzongbo@outlook.com (Z.H.); henry_xscc@mail.xjtu.edu.cn (H.L.); map_bin@163.com (P.M.);
zhangdapeng12@163.com (D.Z.)
\* Correspondence: jiangyu_xian_china@163.com

**Abstract:** Space debris near Earth severely interferes with the development of space, and cataloging space objects is increasingly important. Since optical telescopes and radars used to detect space debris only provide short-arc observations, mathematical algorithms are needed to solve problems in the correlation of observations. In this work, an efficient mathematical algorithm based on $J_2$ analytic solutions is put forward. Initial orbit determination (IOD) serves as the starter and orbit determination (OD) with the weighted least-squares method (WLSM) is used to improve the accuracy of the estimated orbit. Meanwhile, the effect of the weight of different observation types is analyzed. The correlation criteria for bistatic radar observations are accordingly developed. Lastly, the variation in and evolution of the error of bistatic radar ranging are discussed.

**Keywords:** space debris; bistatic radar; correlation; $J_2$ perturbation

## 1. Introduction

The application of space technologies is the theme of this era. Growing uncontrolled space debris and satellites greatly increase the possibility of collisions year by year. The collision of Iridium 33 and Cosmos 2251 is believed to be the first accidental hypervelocity collision of two intact satellites [1]. In August 2016, significant orbit and attitude changes occurred to the Sentinel-1A, which were later proved to be the result of a 1 cm space debris impact [2]. Therefore, there have been many efforts to calculate collision probabilities [3], for collision avoidance [4] and to design space debris removal missions [5]. Above all, cataloging space objects with precise orbits is needed for the good performance of collision avoidance operations and space debris removal missions.

Restricted to the characteristics of current optical or radar surveys for space debris, only short-arc observations, also called as tracklets, can be obtained. If a tracklet can be correlated to one of the cataloged orbits, the tracklet can be used to update the orbit. The left tracklets that are uncorrelated (UCT) are either newly generated debris or an operational satellite after maneuvering. For UCTs, tracklet correlation is usually first needed to accomplish cataloging.

Milani [6] suggested the method of an admissible region (AR) using attributables to solve the observation correlation of asteroids. Tommei and Milani [7] applied the AR method to space debris in Earth orbit and generalized the method to radar cases. Fujimoto [8] gave circular and zero-inclination solutions to the AR method. Farnocchia [9] proposed a virtual debris algorithm based on the AR method. For many correlation works, only two-body integrals were considered. However, Reihs [10] showed that correlation without considering $J_2$ perturbation is effective only when the time interval between two measurements is very limited. Rehis [11] suggested a solution of the AR method considering $J_2$ perturbation.

The Gaussian and Laplacian methods are the most classical IOD algorithms, and they are used in this work. IOD algorithms such as Gibbs are used to deal with radar observations. Details of these algorithms can be found in Escobal [12], Vallado [13], and Liu [14]. Hill [15] took the IOD result as an initial value for an unscented Kalman filter. The calculated orbits and covariances are propagated to a common epoch to accomplish the comparison of orbits. A numerical integral has good precision in propagating orbits and covariances, but the computation requirements are heavy. It is not quite good at dealing with massive amount of tracklets considering the cost of time.

The Air Force Space Surveillance System (AFSSS) has achieved great success in the past few decades because of the wide coverage and its ability to detect high-speed LEO objects. Therefore, continuous attention is being paid to fencelike radar systems. Huang [16] investigated a large-scale distributed space surveillance radar system, and a tracklet association scheme for LEO space debris observed by the double fence radar system was produced [17].

At one step further than other UCTs correlation algorithms, $J_2$ analytic solutions are not only used in orbit calculation but also used in covariance propagation in this work. With $J_2$ analytic solutions, the lack of accuracy by Keplerian integrals can be compensated for, and the cost of time is still much less than that of the numerical integral. For monostatic radars, traditional IOD methods are accurate enough when dealing with a single tracklet. However, the same IOD methods are not quite suitable for the sum of ranges by bistatic radars; thus, an extra OD step is added, and a direct correlation criterion to the observations is raised. The effect of the weight of different observation types in the OD process is analyzed. The criteria can eliminate beforehand outliers that might lead to an error in the correlation process. Lastly, the variation in and evolution of the error of bistatic radar ranging are discussed.

## 2. Initial Orbit Determination for Tracklets Observed by Bistatic Radar

Given the geocentric position $r_T$ of a transmitting station, the geocentric position $r_R$ of the receiving station and the corresponding topocentric position of the space debris $\rho_T$, $\rho_R$, the geocentric position $r$ of space debris can be expressed as $r = \rho_T + r_T$ or $r = \rho_R + r_R$. As shown in Figure 1, there are two types of observation for bistatic radars.

- angles observed by the receiving station, usually azimuth and elevation $(A, E)$ in topocentric horizontal coordinate system for radars;
- the sum of ranges by the transmitting station and receiving station, $\rho = \rho_T + \rho_R$. $\rho_T = \rho_T \hat{\rho}_T$ and $\rho_R = \rho_R \hat{\rho}_R$, $\hat{\rho}_T$ and $\hat{\rho}_R$ are the unit vector of $\rho_T$ and $\rho_R$, $\rho_T$ and $\rho_R$ are the length of $\rho_T$ and $\rho_R$. Sum of ranges $\rho$ can be measured directly, but $\rho_T$ and $\rho_R$ are unknown.

Radar ranging is based on the measurement of signal transmission time, and angles are based on the mechanical measurements of an antenna. Different measurement principles and equipment capabilities result in a difference in accuracy. In normalized units in which the unit of distance is the radius of Earth, and the unit of angles is radian, the error of angles is dozens of times larger than the error of radar ranging.

Traditional IOD methods can deal with monostatic radar ranging, but they are not quite suitable for a sum of ranges by a bistatic radar. Two approximate approaches can be used to obtain an initial orbit from bistatic radar observations.

- Angles only: Since a series of angle observations is sufficient for IOD, a sum of ranges can be temporarily put aside. With Equation (1),

$$\hat{\rho}_R \times r = \hat{\rho}_R \times r_R, \tag{1}$$

  $\rho_R$ from the receiving station to the space debris can be eliminated; therefore, only azimuth and elevation observed by the receiving station are used. The defect of this approach is that low-accuracy measurements are used, and high-accuracy measurements are rejected.

- $\rho_R$ calculation: Since $r_{R2T} = r_T - r_R$, $\hat{\rho}_R$ and $\rho$ are known with a simple geometric calculation, $\rho_R$ can be obtained. Then, the position of space debris can be calculated. The problem of this approach is that the errors of angles are transferred to $\rho_R$.

Since a tracklet is usually short-arc with initial geocentric position and velocity ($r_0 = r(t_0)$, $\dot{r}_0 = \dot{r}(t_0)$) at $t_0$ and prediction duration $\Delta t = t - t_0$, the geocentric position $r(t)$ of the space debris at $t$ could be calculated by series expansion:

$$r(t) = F(r_0, \dot{r}_0, \Delta t)r_0 + G(r_0, \dot{r}_0, \Delta t)\dot{r}_0, \tag{2}$$

where $F$ and $G$ are the polynomial function of $\Delta t$. For the $\rho_R$ calculation approach, Equation (2) has 6 unknown variables ($r_0$, $\dot{r}_0$) and a known vector $\rho_R$. With at least 2 groups of measurements $\rho_R(t_i)(i = 1, 2, 3...)$, ($r_0$, $\dot{r}_0$) would be solvable.

By substituting Equation (2) into Equation (1),

$$\hat{\rho}_R \times (Fr_0 + G\dot{r}_0) = \hat{\rho}_R \times r_R. \tag{3}$$

For the angles-only approach, Equation (3) has 6 unknown variables ($r_0$, $\dot{r}_0$) and 2 known observations $(A, E)$. With at least 3 groups of measurements $\hat{\rho}(t_i)(i = 1, 2, 3...)$, ($r_0$, $\dot{r}_0$) would be solvable.

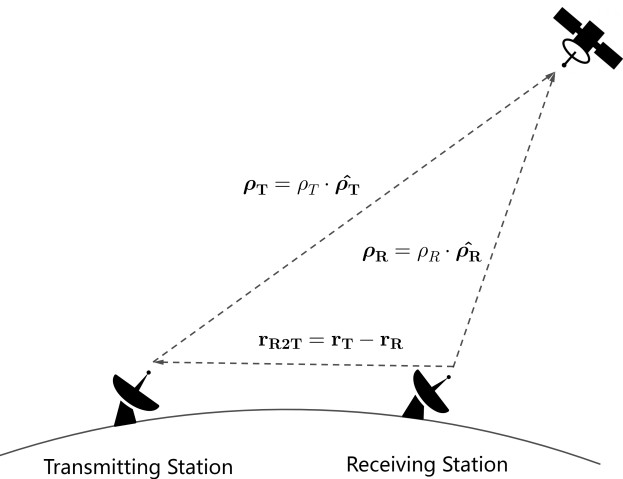

**Figure 1.** Bistatic radar observation.

## 3. Orbit Improvement with Weighted Least Square Method

The angles-only and $\rho_R$ calculation approaches can both provide a set of the estimated state, but the accuracies of the two approaches much depend on the quality of angle measurements. For the angles-only approach, the random noise of $(A, E)$ is directly absorbed by the estimated state. For the $\rho_R$ calculation approach, the random noise of $(A, E)$ is absorbed by $\rho_R$ and also leads to a huge error in the estimated state.

With the weighted least-squares method (WLSM) and an accurate measurement model, the effects of the random noise of measurements can be reduced, measurements with large error can be stripped out, and $\rho$ can be appropriately calculated. Therefore, the accuracy of the orbit can be improved.

### 3.1. Weighted Least-Squares Method

Suppose that $z_i$ is the observation at $t_i$, $x_i$ is the calculated state at $t_i$, and $h(x_i) = h_i(x_0)$ is the observation equation. The loss function is defined as

$$J(x_0) = \sum_{i=1}^{N} \|z_i - h(x_i)\|, \tag{4}$$

the result $x_0^{est}$ should satisfy

$$J(x_0^{est}) = \min_{x_0 \in X_0} J(x_0) \tag{5}$$

where $X_0$ is the state-space of $x_0$. For the least-squares method, either the position and velocity or orbital elements could form the state vector of space debris. For analytic solutions, the state of the space debris is usually expressed by orbital elements.

Define

$$\begin{aligned}
\mathbf{Z} &= (z_1, z_2, ..., z_n)^T, &(6)\\
\mathbf{H}(x_0) &= (h_1(x_0), h_2(x_0), ..., h_n(x_0))^T, &(7)\\
\Delta \mathbf{Z} &= (\Delta z_1, \Delta z_2, ..., \Delta z_n)^T = (z_1 - h_1(x_0), z_2 - h_2(x_0), ..., z_n - h_n(x_0))^T. &(8)
\end{aligned}$$

The loss function can also be expressed as

$$J(x_0) = (\mathbf{Z} - \mathbf{H}(x_0))^T (\mathbf{Z} - \mathbf{H}(x_0)) = \Delta \mathbf{Z}^T \Delta \mathbf{Z}. \tag{9}$$

Supposing that $\Delta x_0 = x_0 - x_0^{true}$ where $x_0^{true}$ is the actual state at $t_0$, we have

$$J(x_0) = \sum_{i=1}^{N} \| \frac{\partial h_i(x_0)}{\partial x_0} \Delta x_0 \|. \tag{10}$$

If $\partial \mathbf{H}(x_0) / \partial \Delta x_0$ is nonsingular, there exists a solution

$$\Delta x_0^{est} = \left[ \left( \frac{\partial \mathbf{H}(x_0)}{\Delta x_0} \right)^T \left( \frac{\partial \mathbf{H}(x_0)}{\Delta x_0} \right) \right]^{-1} \left( \frac{\partial \mathbf{H}(x_0)}{\Delta x_0} \right)^T \Delta \mathbf{Z} \tag{11}$$

which leads to $\partial J(x_0) / \partial \Delta x_0 = 0$.

As mentioned in Section 2, the accuracies of $\rho$ and $(A, E)$ are different. Equal treatment with different types of observations would lower the accuracy of the results, and a proper weight is essential to data fusion. Thus, measurement errors $\sigma_i (i = 1, 2, 3...)$ are put into Equation (11) and form Equation (12).

$$\Delta x_0^{est} = \left[ \left( \frac{\partial \mathbf{H}(x_0)}{\Delta x_0} \right)^T \mathbf{W} \left( \frac{\partial \mathbf{H}(x_0)}{\Delta x_0} \right) \right]^{-1} \left[ \left( \frac{\partial \mathbf{H}(x_0)}{\Delta x_0} \right)^T \mathbf{W} \Delta \mathbf{Z} \right] \tag{12}$$

where

$$\mathbf{W} = \begin{pmatrix}
\sigma_1^{-2} & 0 & ... & 0 & ... & 0 \\
0 & \sigma_2^{-2} & ... & 0 & ... & 0 \\
... & ... & ... & ... & ... & ... \\
0 & 0 & ... & \sigma_i^{-2} & ... & 0 \\
... & ... & ... & ... & ... & ... \\
0 & 0 & ... & 0 & ... & \sigma_n^{-2}
\end{pmatrix}. \tag{13}$$

Since observation equation $h(x_i)$ is an equation with respect to $x_i$, to calculate $\partial h_i(x_0) / \partial x_0$, the state transition matrix $\phi_i$ is needed:

$$\frac{\partial h_i(x_0)}{\partial x_0} = \frac{\partial h(x_i)}{\partial x_i} \phi_i, \tag{14}$$

$$\phi_i = \frac{\partial x_i}{\partial x_0}. \tag{15}$$

### 3.2. Effect of Weight

Theoretical weight in WLSM is the accuracy of observation, as shown in Equation (13). In practice, it is not easy to obtain the exact error of each observation while tracking and

observing. The errors of observations are different since the error is affected by multiple factors. Usually, a composite weight from experience and equipment performance is set for each type of observation.

In this work, the relative weight between different types of observations could affect the accuracy of a certain orbital element. Two stations, as shown in Table 1, and a satellite, as shown in Table 2, were simulated to demonstrate the effect of observation weight. The duration of the simulated tracklet was 1 min.

**Table 1.** Description of two simulated stations.

| Simulated Stations | Latitude (deg) | Longitude (deg) | Height (m) |
|---|---|---|---|
| Transmitting station | 30 | 108 | 0 |
| Receiving station | 30 | 105 | 0 |

**Table 2.** Description of the simulated satellite.

| Semi-Major Axis (m) | Eccentricity | Inclination (deg) | RAAN (deg) |
|---|---|---|---|
| 6,878,137.0 | 0.001 | 60.0 | 60.0 |

As discussed in Section 2, there are mainly two types of observation for bistatic radar, and the accuracy of radar ranging is usually better than that of angles. From Equation (12), the results of estimation change with the relative weight between different observations, instead of the absolute weight of observations. In normalized units which the unit of distance is the radius of Earth, and the unit of angles is radian, variation in the relative error between the sum of ranges and azimuth ($\sigma_\rho/\sigma_A$) was set from 0.1 to 0.02, and variation in the relative error between elevation and azimuth ($\sigma_E/\sigma_A$) was set from 0.5 to 2.0. Like the relative errors of observations being used to describe weight, the relative error between different orbital elements is used to describe the accuracy of certain orbital element. By setting the error of the estimated eccentricity as the reference, the relative error between the semimajor axis and eccentricity ($\sigma_a/\sigma_e$) can represent the accuracy of the estimated semimajor axis, and the relative error between inclination and eccentricity ($\sigma_i/\sigma_e$) can represent the accuracy of the estimated inclination. The effects of weight on the accuracy of estimated orbital elements are shown in Figure 2.

Figure 2 shows that the accuracy of the estimated semimajor axis increases with the weight of radar ranging. This effect becomes stronger when $\sigma_E \neq \sigma_A$. On the other hand, the accuracy of inclination decreases with the weight of radar ranging, and increases with the weight of elevation.

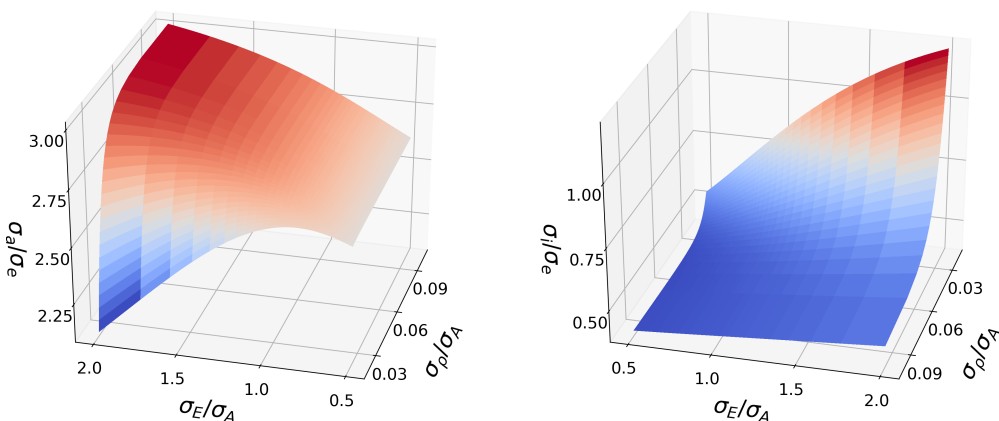

**Figure 2.** Effect of weight on the accuracy of estimated orbital elements.

## 4. Correlation Considering $J_2$ Perturbation

The motion of space debris in terrestrial space is affected by all kinds of perturbations, such as drag, solar radiation pressure, and the gravitational perturbations of the Sun and Moon. Among all, the $J_2$ term of Earth's nonspherical perturbation has the strongest influence.

The $J_2$ term represents the perturbation caused by the oblateness of Earth. The acceleration of $J_2$ perturbation is shown in Equation (16),

$$
\ddot{\boldsymbol{r}}_{J2} = \begin{bmatrix} -\frac{3}{2}J_2 \frac{GM_e R_e^2}{r_{ecf}^5}\left(1 - 5\frac{z_{ecf}^2}{r_{ecf}^2}\right)x_{ecf} \\[2mm] -\frac{3}{2}J_2 \frac{GM_e R_e^2}{r_{ecf}^5}\left(1 - 5\frac{z_{ecf}^2}{r_{ecf}^2}\right)y_{ecf} \\[2mm] -\frac{3}{2}J_2 \frac{GM_e R_e^2}{r_{ecf}^5}\left(3 - 5\frac{z_{ecf}^2}{r_{ecf}^2}\right)z_{ecf} \end{bmatrix}.
\tag{16}
$$

$G$ is the gravitational constant, $M_e$ is the mass of Earth, $R_e$ is the radius of Earth, and $\boldsymbol{r}_{ecf} = (x_{ecf}, y_{ecf}, z_{ecf})$ is the position of space debris in an Earth-centered fixed coordinate system. $J_2$ perturbation is not only considered in orbit prediction, but also in covariance propagation in this work.

### 4.1. Orbit Propagation with $J_2$ Perturbation

$J_2$ perturbation causes a secular variation in the orbital plane (right ascension of ascending node, $\Omega$) and argument of perigee ($\omega$) as shown in Equations (17) and (18):

$$
\dot{\Omega} = -\frac{3}{2}J_2\left(\frac{R_e}{p}\right)^2 n \cos i
\tag{17}
$$

$$
\dot{\omega} = \frac{3}{2}J_2\left(\frac{R_e}{p}\right)^2 n\left(2 - \frac{5}{2}\sin^2 i\right)
\tag{18}
$$

where $R_e$ is the radius of Earth, $n$ is the mean motion of a satellite, and $p = a(1 - e^2)$. Equation (19) shows that the mean motion ($n$) of space debris is affected by $J_2$ perturbation, but has no secular variation:

$$
n_{J_2} = n + \frac{3}{4}J_2\left(\frac{R_e}{a}\right)^2\left(\frac{2 - 3\sin^2 i}{p^{\frac{3}{2}}}\right).
\tag{19}
$$

Since $M = M_0 + n(t - t_0)$, variation in $n$ leads to extra secular variation in mean anomaly ($M$). Equation (20) gives the expression of $\dot{M}$:

$$
\dot{M} = n + \frac{3}{2}nJ_2\left(\frac{R_e}{p}\right)^2\left(1 - \frac{3}{2}\sin^2 i\right)\sqrt{1 - e^2}.
\tag{20}
$$

With Equations (17)–(20), analytic state transition matrix $\boldsymbol{\phi} = \boldsymbol{\phi}^{(0)} + \boldsymbol{\phi}^{(1)}$,

$$\boldsymbol{\phi}^{(0)} = \begin{pmatrix} 1 & 0 & 0 & 0 & 0 & 0 \\ 0 & 1 & 0 & 0 & 0 & 0 \\ 0 & 0 & 1 & 0 & 0 & 0 \\ 0 & 0 & 0 & 1 & 0 & 0 \\ 0 & 0 & 0 & 0 & 1 & 0 \\ \phi_{61}^{(0)} & 0 & 0 & 0 & 0 & 1 \end{pmatrix}, \tag{21}$$

$$\boldsymbol{\phi}^{(1)} = \begin{pmatrix} 0 & 0 & 0 & 0 & 0 & 0 \\ 0 & 0 & 0 & 0 & 0 & 0 \\ 0 & 0 & 0 & 0 & 0 & 0 \\ \phi_{41}^{(1)} & \phi_{42}^{(1)} & \phi_{43}^{(1)} & 0 & 0 & 0 \\ \phi_{51}^{(1)} & \phi_{52}^{(1)} & \phi_{53}^{(1)} & 0 & 0 & 0 \\ \phi_{61}^{(1)} & \phi_{62}^{(1)} & \phi_{63}^{(1)} & 0 & 0 & 0 \end{pmatrix}, \tag{22}$$

where $\phi_{61}^{(0)}$ is a function of $(a_0, \Delta t)$, and $\phi_{ij}^{(1)}$ are functions of $(a_0, e_0, i_0, \Delta t)$. $\boldsymbol{\phi}^{(0)}$ represents the state transition of Keplerian motion, and $\boldsymbol{\phi}^{(1)}$ represents the effect of $J_2$ perturbation.

*4.2. Correlation of Tracklets*

Assuming that there are $n$ tracklets $(1, 2, ...j, ..., k, ..., n)$, and each tracklet has more than 3 groups of measurements, the correlation between the $j$th tracklet and one of the measurements in the $k$th tracklet is taken as an example.

After IOD and OD with WLSM, an improved state $(x_j)$ could be obtained for the $j$th tracklet. Propagating $x_j$ and the error of $x_j$ to the $k$th tracklet, observation $(A_k^j, E_k^j, \rho_k^j)$ and the error of observation $(\Delta A_k^j, \Delta E_k^j, \Delta \rho_k^j)$ can be calculated. The error of $x_j$ by OD with only one tracklet was much smaller than the actual deviation of $x_j$. In order to more accurately calculate $(\Delta A_k^j, \Delta E_k^j, \Delta \rho_k^j)$, the empirical error of the estimated orbit with one tracklet is needed.

Since the observation error $(\Delta A, \Delta E, \Delta \rho_R)$ of the receiving station was pairwise orthogonal, $(A, E, \rho_R)$ should conform to the restriction of the error ellipsoid as shown in Figure 3.

If the $j$th and $k$th tracklets belong to the same satellite, $(A, E, \rho_R)_k$ should satisfy Equation (23):

$$\left( \frac{A_k - A_k^j}{m\Delta A_k^j} \right)^2 + \left( \frac{E_k - E_k^j}{m\Delta E_k^j} \right)^2 + \left( \frac{(\rho_R)_k - (\rho_R)_k^j}{m(\Delta \rho_R)_k^j} \right)^2 < 1, \tag{23}$$

$m$ is the coefficient absorbing the inaccuracy of dynamic models and the error growth in propagation.

The error of bistatic radar ranging $(\Delta \rho)$ is affected by three factors:

$\Delta \rho_R$ the error produced by the receiving station;

$\Delta \rho_T$ the error produced by the transmitting station;

$\Delta \rho_s$ the systematic error between the receiving station and the transmitting station.

$$\Delta \rho = f(\Delta \rho_R, \Delta \rho_T, \Delta \rho_s), \tag{24}$$

$\Delta \rho_s$ is mainly decided by the performance of time synchronization between different stations. According to Guo [18], timing with a global navigation satellite system (GNSS) is about 0.1 ns. Thus, $\Delta \rho_s \ll \Delta \rho_R, \Delta \rho_T$ at present. Assuming that the variation in $\Delta \rho_R$

and $\Delta\rho_T$ is mainly affected by $J_2$ perturbation, and Equation (24) can be approximated to Equation (25) in one tracklet:

$$\frac{\rho_R}{\rho} \cong \frac{\Delta\rho_R}{\Delta\rho}, \tag{25}$$

Equation (23) can be transformed into Equation (26),

$$\left(\frac{A_k - A_k^j}{m\Delta A_k^j}\right)^2 + \left(\frac{E_k - E_k^j}{m\Delta E_k^j}\right)^2 + \left(\frac{\rho_k - \rho_k^j}{m\Delta\rho_k^j}\right)^2 < 1. \tag{26}$$

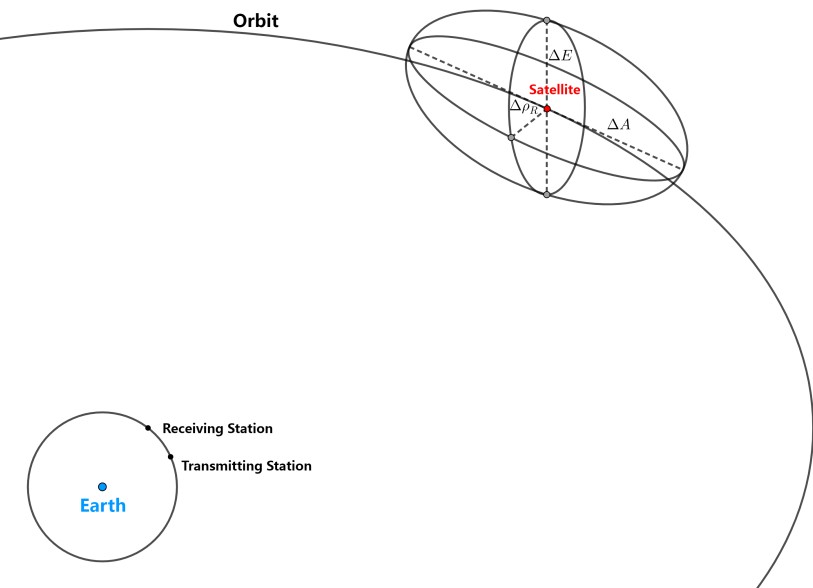

**Figure 3.** Error ellipsoid of observations of a receiving station.

Different from calculating the Mahalanobis distance of two orbits [11,19], each group of observations was tested with Equation (26). If 70% measurements of the tracklet were successfully correlated, the tracklet was successfully correlated. For tracklets with clean data, the effect of the proposed approach is similar to that of calculating Mahalanobis distance. However, tracklets with mixed measurements of different space debris appear now and then, and mixed measurements could lead to a failure in an OD process or an estimated orbit with huge error. With the proposed approach, correlation and data cleansing can be accomplished in one step.

There were quite a few miscorrelations only with Equation (26). Orbit determination with WLSM can also be used to screen out miscorrelations. Two tracklets were insufficient for confirmation. Tests with real data were reported by Tommei [20], who found that the correctness of correlation would be largely increased when at least 3 tracklets are confirmed by the least-square method. In this work, only correlated observations, instead of the entire tracklet, were used to implement the confirmation.

## 5. Discussion

Since the properties of azimuth and elevation observed by a receiving station are the same with those observed by monostatic radars, which was discussed by Cordelli [21], $\Delta A$ and $\Delta E$ are not discussed in the following. Two issues are discussed in detail:

1. the performance of orbit determination with WLSM for a single bistatic radar tracklet;
2. the effect of orbital elements' accuracy and prediction duration on $\Delta\rho$.

### 5.1. Accuracy of Orbit Determination

In order to test the performance of orbit determination, 4298 tracklets of 3538 LEO satellites were simulated. Detailed simulation strategies are shown in Table 3.

**Table 3.** Strategies of simulation.

| Subject | Content |
|---|---|
| Orbital elements | Two-Line-Element (TLE) |
| Dynamic model | sgp4 |
| Minimal height threshold | 200 km |
| Maximal height threshold | 1700 km |
| Minimal time span | 20 s |
| Maximal time span | 300 s |
| Mean time span | 120 s |
| $\sigma$ of azimuth noise | 0.1° |
| $\sigma$ of elevation noise | 0.1° |
| $\sigma$ of $\rho$ noise | 50 m |

For the initial orbit determination demonstration, the $\rho_R$ calculation approach was selected. Figure 4 gives the deviation between the estimated orbit elements by IOD and the true orbital elements (TLE). $\mu$ is the mathematical expectation of the deviation which represents the systematic bias of the estimated orbital elements.

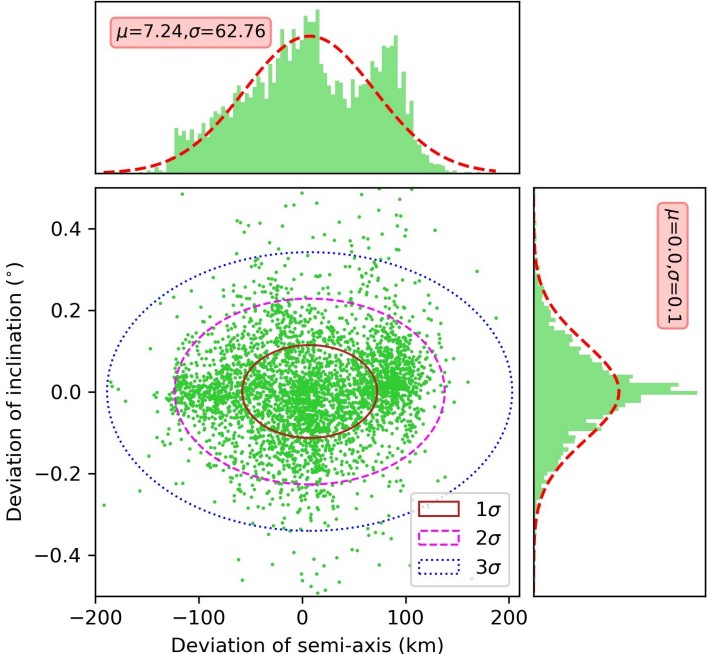

**Figure 4.** Accuracy of $\rho_R$ calculation approach.

The estimated inclination by IOD barely had systematic bias, and the error reached $\sigma \sim 0.1°$. On the other hand, the systematic bias of the estimated semimajor axis by IOD was as large as several kilometers, and the standard deviation was even larger.

Figure 5 gives the deviation between the estimated orbital elements by WLSM and the true orbital elements (TLE).

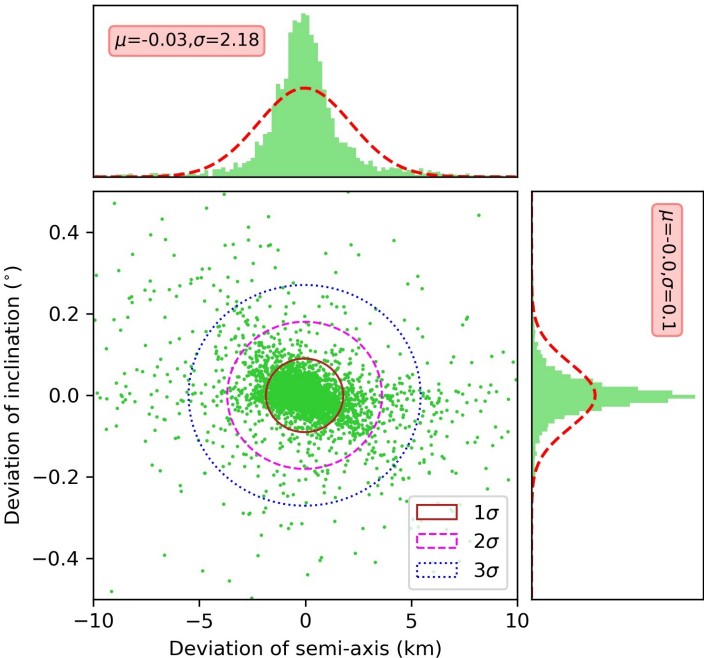

**Figure 5.** Accuracy of orbit determination with WLSM.

The accuracy of the estimated inclination by OD was only slightly higher than that of the estimated inclination by IOD. However, the estimated semimajor axis was significantly improved by OD with WLSM. The systematic bias of the estimated semimajor axis dropped to the order of magnitude of ten meters, and the accuracy of the estimated semimajor axis increased several dozens of times. This improvement was largely due to the proper use of $\rho$. $\rho$ had a strong restriction on the estimation of the semimajor axis, and this phenomenon corresponded to the effect of weight discussed in Section 3.2.

Two more things were also noticed from the results of orbit determination:

- Orbit improvement with WLSM is indispensable. Since the error of the estimated orbit elements by IOD was too large, the number of miscorrelations would grow rapidly as the interval between tracklets increased. At the same time, the systematic bias of the estimated semimajor axis would render the orbit propagation wrong.
- As mentioned in Section 4.2, an empirical error of the estimated orbit is needed to calculate $(\Delta A_k^j, \Delta E_k^j, \Delta\rho_k^j)$ in Equation (26). From Figure 5, the empirical error can be obtained.

### 5.2. Variation in and Evolution of $\Delta\rho$

$\Delta\rho$ is mainly affected by the accuracy of orbit determination and the prediction duration. In order to test the effect of different factors, the two stations in Table 1 and the satellite in Table 2 were chosen to accomplish the experiment. Assuming that the prediction duration was 1 day, Figure 6 shows the variation in $\Delta\rho$ with respect to the estimated orbit element and its accuracy.

$\Delta\rho$ was easily found to always be positively associated with the absolute error of an orbital element. This feature can substantially simplify the calculation because the extreme value is sufficient for calculating the confidence zone of $\Delta\rho$ instead of traversing all possible errors of orbital elements. Figure 6 also shows that $\Delta\rho$ became smaller for the satellites of a higher altitude with the same semimajor axis error. The effect of the right ascension of the ascending node ($\Omega$) was almost identical to that of inclination, which demonstrates that the orbit plane had no direct relationship with $\Delta\rho$.

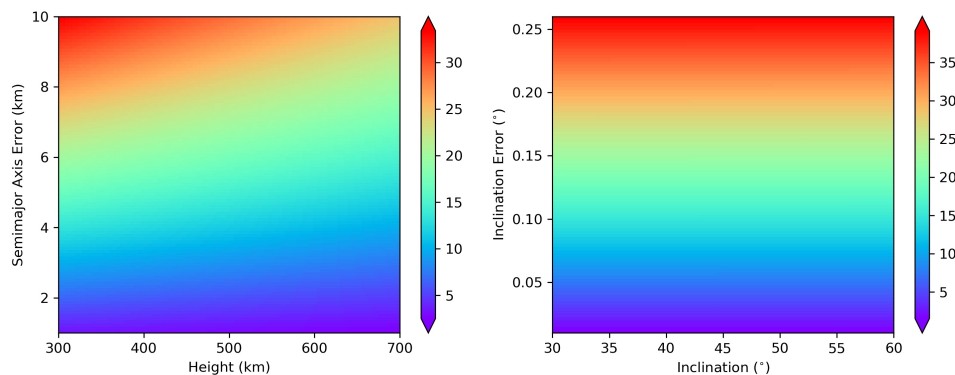

**Figure 6.** $\Delta\rho$ with respect to the accuracy of the estimated orbital elements.

Assuming that the orbit was calculated with the tracklet in Pass 0, the orbit and its error propagated to tracklets in Passes 1, 10, and 14. The evolution of $\rho$ and $\Delta\rho$ is shown in Figure 7.

Figure 7 shows that $\rho$ in one tracklet could vary by several thousand kilometers for a satellite with an altitude of 500 km, while $\Delta\rho$ only varies little. This shows that $\Delta\rho$ barely had a relationship with $\rho$. $\sigma_\rho$ is not always positively associated with prediction duration. If $\sigma_\rho$ drops, either the error of velocity or the error of azimuth and elevation grows.

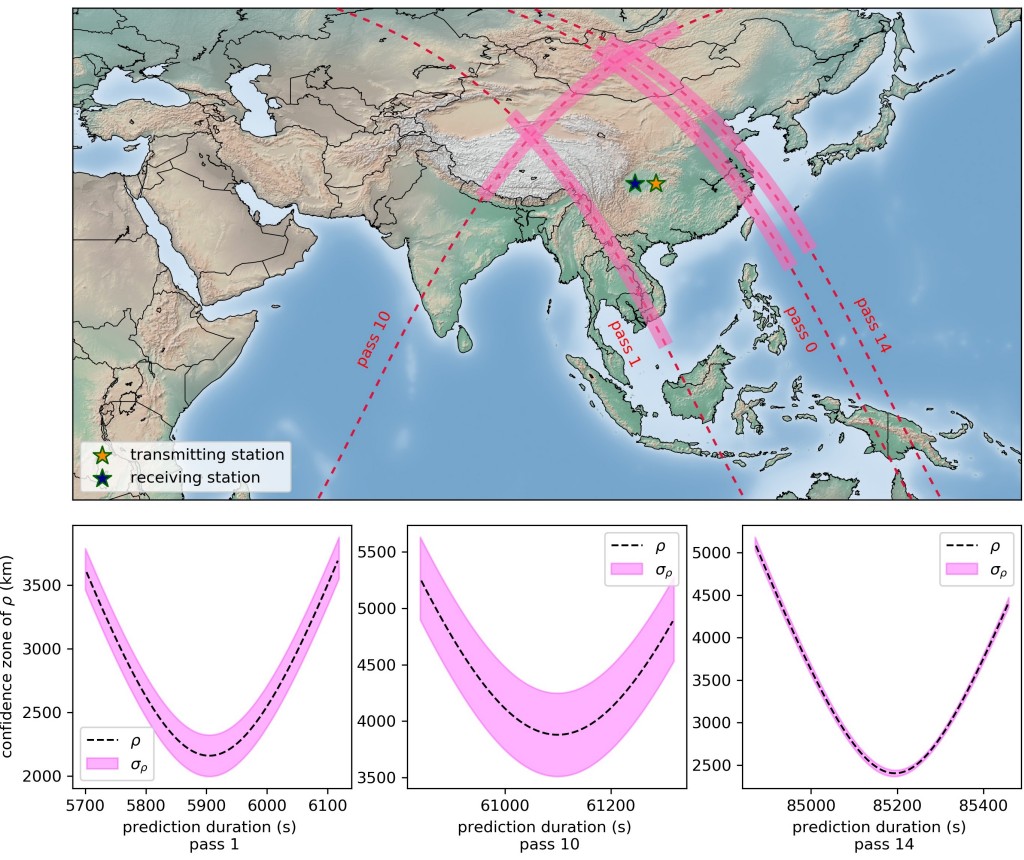

**Figure 7.** Evolution of $\rho$ and $\Delta\rho$.

## 6. Conclusions

In this work, an efficient algorithm was presented to deal with the UCT correlation problem. The algorithm was based on $J_2$ analytic solutions for orbit and covariance propagation. The lack of accuracy of Keplerian integral can be compensated to a certain level by taking $J_2$ perturbation into consideration.

The process of correlation starts with the IOD of a tracklet, followed by obtaining an improved orbit with WLSM. An empirical error of the estimated state is used to form the covariance. The OD with an analytic orbit and covariance propagation runs fast for sparse data, which also significantly decreases the systematic bias of the estimated semimajor axis, and the accuracy of the estimated semimajor axis increases several dozens of times. The orbit and covariance are propagated to the epoch of the second tracklet, and Equation (26) was used to perform the correlation. Instead of OD for the second tracklet and comparing the estimated orbit, each pair of observations in the second tracklet were separately correlated. If 70% observations of the tracklet were successfully correlated, the tracklet was successfully correlated. With the proposed approach, correlation and data cleansing can be accomplished in one step. However, only the correlated observations in the tracklet are used in the next step to implement the confirmation, and update the orbit and covariance. The accuracy of the semimajor axis increased with the weight of radar ranging. This effect became stronger when $\sigma_E \neq \sigma_A$. On the other hand, the accuracy of inclination decreased with the weight of radar ranging, and increased with the weight of elevation. The error of bistatic radar ranging also became smaller for space debris of higher altitude with the same semimajor axis error, and the orbit plane had no direct relationship with the error of bistatic radar ranging.

**Author Contributions:** Conceptualization, Z.H. and P.M.; Data curation, D.Z.; Formal analysis, H.L.; Funding acquisition, Y.J.; Investigation, Z.H., Y.J. and H.L.; Methodology, Z.H. and Y.J.; Project administration, Y.J. and H.L.; Resources, Y.J.; Software, Z.H. and D.Z.; Supervision, H.L.; Validation, Y.J. and P.M.; Visualization, Z.H.; Writing—original draft, Z.H.; Writing—review & editing, Z.H., Y.J., H.L. and P.M. All authors have read and agreed to the published version of the manuscript.

**Funding:** This research was funded by National Natural Science Foundation of China grant number U21B2050.

**Conflicts of Interest:** The authors declare no conflict of interest.

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
