# Peer review of "Bistatic Radar Observations Correlation of LEO Satellites Considering J2 Perturbation"

_mathematics, doi:10.3390/math10132197_

Round 1

Reviewer 1 Report

The paper can be interesting for many readers, but the presentation of the problem and solution needs improvement.

First, the authors claim in the introduction that "In this work, J 2 analytic solutions for orbit and covariance propagation are derived
respectively". Then, I would expect them to denote clearly which part of the equations presented are the well-known relationships from the literature and which equations are derived by the authors.
Also, the claim that "solutions are derived" suggests that the derivation is shown in the paper (and not only the result).

Second, the equations are not explained well enough for a general audience. Starting from eq 1, where the meaning of "hat" is not defined, through state-space equations without specifying which the state variables form the state vector and how they are denoted, to eq 22, where some strange variable m appears (and is not explained until eq 25 on the next page). There are many other poorly defined variables - maybe the notation is not a problem for specialists in OD, but the wider audience would need the clarification.

Third, it is not shown clearly how the discussed problem is specific to a bistatic radar. I would say that the only remark is around eq 23, and in general, the problem is reduced to one identical with a monostatic setup.

Some minor remarks:

  • Fig 2 needs a better explanation of axes, and the labels are too small
  • I suggest one paragraph explaining what the J2 term is in practice
  • I would welcome a broader explanation and comparison of the ideas presented here with the concept of Mahalanobis distance (line 154); they seem pretty similar.
  • "The accuracy of ranging is much better than angles" (line 66) - a practical example is welcome here

Reviewer 2 Report

This paper proposes a correlation algorithm to handle uncorrelated (UCT) objects (newly generated debris or maneuvering LEO satellite) under a bistatic radar observations scenario. The main contribution of this paper is considering the J2 perturbation in the correlation algorithm. The algorithm starts with the Initial orbit determination (IOD) of a tracklet and followed by Weighted Least Square Method to improve the accuracy. The literature review is sufficient. However, there are several issues to be addressed before accept for publication.

Major issues:

  1. pp. 3, line 69-71, the inputs of both angles only and ρcalculation methods should be clarified. For example, line 65 explicitly mentions both ρand ρare unknown. While line 71 mentions "... ρ are known", which ρ is known?
  2. pp. 3, line 89, the state space of xi should be clarified at the beginning to remove potential confusion for readers. The general state-space of orbital mechanics variables includes orbital elements and r/v vectors that both are widely used in the literature.
  3. pp. 10, line 214-216, the result analysis is not convincing. "It shows that Δρ barely has relationship with ρ. Although, ...". Fig. 7 alone cannot provide us with this conclusion. Based on the two data points in Fig. 7 only, it's totally fine to say Δρ or σρ is positively associated with the ρ or the prediction duration. More justification or data is needed.

Minor issues:

  1. pp. 2, Fig. 1, the direction of the arrows is not correct. Both ρand ρshould point from the radar station to the satellite.
  2. pp. 5, line 117-118, please specify the symbol representations for σa and σi in the main context. They appear in the figure with no explanations.
  3. pp. 8, line 162, "conformation" or "confirmation"?

Round 2

Reviewer 1 Report

All my questions have been answered in the new revision.